# Empirical Bayes Transductive Meta-Learning with Synthetic Gradients

**Shell Xu Hu**[1]   **Pablo G. Moreno**[2]   **Yang Xiao**[1]   **Xi Shen**[1]
**Guillaume Obozinski**[3]   **Neil D. Lawrence**[4]   **Andreas Damianou**[2]

[1]**École des Ponts ParisTech**
Champs-sur-Marne, France
{xu.hu, yang.xiao, xi.shen}@enpc.fr

[2]**Amazon**
Cambridge, United Kingdom
{morepabl, damianou}@amazon.com

[3]**Swiss Data Science Center**
Lausanne, Switzerland
guillaume.obozinski@epfl.ch

[4]**University of Cambridge**
Cambridge, United Kingdom
ndl21@cam.ac.uk

## Abstract

We propose a meta-learning approach that learns from multiple tasks in a transductive setting, by leveraging the unlabeled query set in addition to the support set to generate a more powerful model for each task. To develop our framework, we revisit the empirical Bayes formulation for multi-task learning. The evidence lower bound of the marginal log-likelihood of empirical Bayes decomposes as a sum of local KL divergences between the variational posterior and the true posterior on the query set of each task. We derive a novel amortized variational inference that couples all the variational posteriors via a meta-model, which consists of a synthetic gradient network and an initialization network. Each variational posterior is derived from synthetic gradient descent to approximate the true posterior on the query set, although where we do not have access to the true gradient. Our results on the Mini-ImageNet and CIFAR-FS benchmarks for episodic few-shot classification outperform previous state-of-the-art methods. Besides, we conduct two zero-shot learning experiments to further explore the potential of the synthetic gradient.

## 1 Introduction

While supervised learning of deep neural networks can achieve or even surpass human-level performance (He et al., 2015; Devlin et al., 2018), they can hardly extrapolate the learned knowledge beyond the domain where the supervision is provided. The problem of solving rapidly a new task after learning several other similar tasks is called *meta-learning* (Schmidhuber, 1987; Bengio et al., 1991; Thrun & Pratt, 1998); typically, the data is presented in a two-level hierarchy such that each data point at the higher level is itself a dataset associated with a task, and the goal is to learn a *meta-model* that generalizes across tasks. In this paper, we mainly focus on *few-shot learning* (Vinyals et al., 2016), an instance of meta-learning problems, where a task $t$ consists of a *query set* $d_t := \{(x_{t,i}, y_{t,i})\}_{i=1}^n$ serving as the test-set of the task and a *support set* $d_t^l := \{(x_{t,i}^l, y_{t,i}^l)\}_{i=1}^{n^l}$ serving as the train-set. In *meta-testing*[1], one is given the support set and the inputs of the query set $x_t := \{x_{t,i}\}_{i=1}^n$, and asked to predict the labels $y_t := \{y_{t,i}\}_{i=1}^n$. In *meta-training*, $y_t$ is provided as the ground truth. The setup of few-shot learning is summarized in Table 1.

A important distinction to make is whether a task is solved in a *transductive* or *inductive* manner, that is, whether $x_t$ is used. The inductive setting is what was originally proposed by Vinyals et al. (2016), in which only $d_t^l$ is used to generate a model. The transductive setting, as an alternative, has the advantage of being able to see partial or all points in $x_t$ before making predictions. In fact,

---

[1]To distinguish from testing and training within a task, meta-testing and meta-training are referred to as testing and training over tasks.

|  | **Support set** | **Query set** |  |
|---|---|---|---|
|  | $d_t^l := \{(x_{t,i}^l, y_{t,i}^l)\}_{i=1}^{n^l}$ | $x_t := \{x_{t,i}\}_{i=1}^{n}$ | $y_t = \{y_{t,i}\}_{i=1}^{n}$ |
| Meta-training | ✓ | ✓ | ✓ |
| Meta-testing | ✓ | ✓ | ✗ |

Table 1: The setup of few-shot learning. If task $t$ is used for meta-testing, $y_t$ is not given to the model.

Nichol et al. (2018) notice that most of the existing meta-learning methods involve transduction unintentionally since they use $x_t$ implicitly via the *batch normalization* (Ioffe & Szegedy, 2015). Explicit transduction is less explored in meta-learning, the exception is Liu et al. (2018), who adapted the idea of label propagation (Zhu et al., 2003) from graph-based semi-supervised learning methods. We take a totally different path that meta-learn the "gradient" descent on $x_t$ without using $y_t$.

Due to the hierarchical structure of the data, it is natural to formulate meta-learning by a *hierarchical Bayes* (HB) model (Good, 1980; Berger, 1985), or alternatively, an empirical Bayes (EB) model (Robbins, 1985; Kucukelbir & Blei, 2014). The difference is that the latter restricts the learning of meta-parameters to point estimates. In this paper, we focus on the EB model, as it largely simplifies the training and testing without losing the strength of the HB formulation.

The idea of using HB or EB for meta-learning is not new: Amit & Meir (2018) derive an objective similar to that of HB using PAC-Bayesian analysis; Grant et al. (2018) show that MAML (Finn et al., 2017) can be understood as a EB method; Ravi & Beatson (2018) consider a HB extension to MAML and compute posteriors via amortized variational inference. However, unlike our proposal, these methods do not make full use of the unlabeled data in query set. Roughly speaking, they construct the variational posterior as a function of the labeled set $d_t^l$ without taking advantage of the unlabeled set $x_t$. The situation is similar in gradient based meta-learning methods (Finn et al., 2017; Ravi & Larochelle, 2016; Li et al., 2017b; Nichol et al., 2018; Flennerhag et al., 2019) and many other meta-learning methods (Vinyals et al., 2016; Snell et al., 2017; Gidaris & Komodakis, 2018), where the mechanisms used to generate the task-specific parameters rely on groundtruth labels, thus, there is no place for the unlabeled set to contribute. We argue that this is a suboptimal choice, which may lead to overfitting when the labeled set is small and hinder the possibility of zero-shot learning (when the labeled set is not provided).

In this paper, we propose to use synthetic gradient (Jaderberg et al., 2017) to enable transduction, such that the variational posterior is implemented as a function of the labeled set $d_t^l$ and the unlabeled set $x_t$. The synthetic gradient is produced by chaining the output of a gradient network into auto-differentiation, which yields a surrogate of the inaccessible true gradient. The optimization process is similar to the inner gradient descent in MAML, but it iterates on the unlabeled $x_t$ rather than on the labeled $d_t^l$, since it does not rely on $y_t$ to compute the true gradient. The labeled set for generating the model for an unseen task is now optional, which is only used to compute the initialization of model weights in our case. In summary, our main contributions are the following:

1. In section 2 and section 3, we develop a novel empirical Bayes formulation with transduction for meta-learning. To perform amortized variational inference, we propose a parameterization for the variational posterior based on synthetic gradient descent, which incoporates the contextual information from all the inputs of the query set.

2. In section 4, we show in theory that a transductive variational posterior yields better generalization performance. The generalization analysis is done through the connection between empirical Bayes formulation and a multitask extension of the information bottleneck principle. In light of this, we name our method *synthetic information bottleneck* (SIB).

3. In section 5, we verify our proposal empirically. Our experimental results demonstrate that our method significantly outperforms the state-of-the-art meta-learning methods on few-shot classification benchmarks under the one-shot setting.

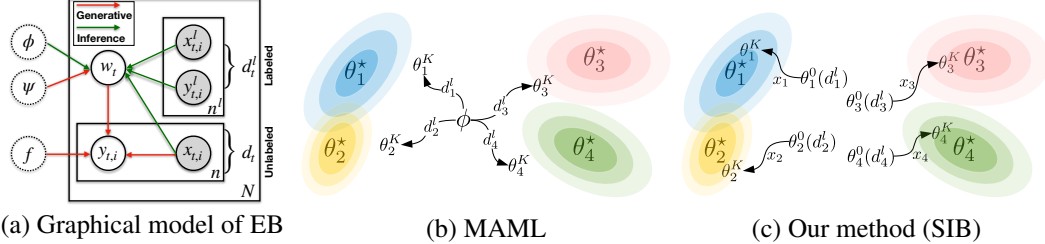

(a) Graphical model of EB      (b) MAML      (c) Our method (SIB)

Figure 1: **(a)** The generative and inference processes of the empirical Bayes model are depicted in solid and dashed arrows respectively, where the meta-parameters are denoted by dashed circles due to the point estimates. A comparison between MAML (6) and our method (SIB) (10) is shown in **(b)** and **(c)**. MAML is an inductive method since, for a task $t$, it first constructs the variational posterior (with parameter $\theta^K$) as a function of the support set $d_t^l$, and then test on the unlabeled $x_t$; while SIB uses a better variational posterior as a function of both $d_t^l$ and $x_t$: it starts from an initialization $\theta_t^0(d_t^l)$ generated using $d_t^l$, and then yields $\theta_t^K$ by running $K$ synthetic gradient steps on $x_t$.

## 2 META-LEARNING WITH TRANSDUCTIVE INFERENCE

The goal of meta-learning is to train a *meta-model* on a collection of tasks, such that it works well on another disjoint collection of tasks. Suppose that we are given a collection of $N$ tasks for training. The associated data is denoted by $\mathcal{D} := \{d_t := (x_t, y_t)\}_{t=1}^N$. In the case of few-shot learning, we are given in addition a support set $d_t^l$ in each task. In this section, we revisit the classical empirical Bayes model for meta-learning. Then, we propose to use a transductive scheme in the variational inference by implementing the variational posterior as a function of $x_t$.

### 2.1 EMPIRICAL BAYES MODEL

Due to the hierarchical structure among data, it is natural to consider a hierarchical Bayes model with the marginal likelihood

$$p_f(\mathcal{D}) = \int_\psi p_f(\mathcal{D}|\psi)p(\psi) = \int_\psi \Big[\prod_{t=1}^N \int_{w_t} p_f(d_t|w_t)p(w_t|\psi)\Big]p(\psi). \tag{1}$$

The generative process is illustrated in Figure 1 (a, in red arrows): first, a *meta-parameter* $\psi$ (i.e., hyper-parameter) is sampled from the *hyper-prior* $p(\psi)$; then, for each task, a *task-specific parameter* $w_t$ is sampled from the *prior* $p(w_t|\psi)$; finally, the dataset is drawn from the *likelihood* $p_f(d_t|w_t)$. Without loss of generality, we assume the log-likelihood model factorizes as

$$\log p_f(d_t|w_t) = \sum_{i=1}^n \log p_f(y_{t,i}|x_{t,i}, w_t) + \log p(x_{t,i}|w_t),$$

$$= \sum_{i=1}^n -\frac{1}{n}\ell_t\big(\hat{y}_{t,i}(f(x_{t,i}), w_t), y_{t,i}\big) + \text{ constant}. \tag{2}$$

It is the setting advocated by Minka (2005), in which the generative model $p(x_{t,i}|w_t)$ can be safely ignored since it is irrelevant to the prediction of $y_t$. To simplify the presentation, we still keep the notation $p_f(d_t|w_t)$ for the likelihood of the task $t$ and use $\ell_t$ to specify the discriminative model, which is also referred to as the *task-specific loss*, e.g., the cross entropy loss. The first argument in $\ell_t$ is the prediction, denoted by $\hat{y}_{t,i} = \hat{y}_{t,i}(f(x_{t,i}), w_t)$, which depends on the *feature representation* $f(x_{t,i})$ and the *task-specific weight* $w_t$.

Note that rather than following a fully Bayesian approach, we leave some random variables to be estimated in a frequentist way, e.g., $f$ is a *meta-parameter* of the likelihood model shared by all tasks, for which we use a point estimate. As such, the posterior inference about these variables will be largely simplified. For the same reason, we derive the *empirical Bayes* (Robbins, 1985; Kucukelbir & Blei, 2014) by taking a point estimate on $\psi$. The marginal likelihood

now reads as

$$p_{\psi,f}(\mathcal{D}) = \prod_{t=1}^{N} \int_{w_t} p_f(d_t|w_t) p_\psi(w_t). \tag{3}$$

We highlight the meta-parameters as subscripts of the corresponding distributions to distinguish from random variables. Indeed, we are not the first to formulate meta-learning as empirical Bayes. The overall model formulation is essentially the same as the ones considered by Amit & Meir (2018); Grant et al. (2018); Ravi & Beatson (2018). Our contribution lies in the variational inference for empirical Bayes.

## 2.2 AMORTIZED INFERENCE WITH TRANSDUCTION

As in standard probabilistic modeling, we derive an *evidence lower bound* (ELBO) on the log version of (3) by introducing a variational distribution $q_{\theta_t}(w_t)$ for each task with parameter $\theta_t$:

$$\log p_{\psi,f}(\mathcal{D}) \geq \sum_{t=1}^{N} \Big[ \mathbb{E}_{w_t \sim q_{\theta_t}} \big[ \log p_f(d_t|w_t) \big] - D_{\mathrm{KL}}\big( q_{\theta_t}(w_t) \| p_\psi(w_t) \big) \Big]. \tag{4}$$

The variational inference amounts to maximizing the ELBO with respect to $\theta_1, \ldots, \theta_N$, which together with the maximum likelihood estimation of the meta-parameters, we have the following optimization problem:

$$\min_{\psi,f} \min_{\theta_1,\ldots,\theta_N} \frac{1}{N} \sum_{t=1}^{N} \Big[ \mathbb{E}_{w_t \sim q_{\theta_t}} \big[ -\log p_f(d_t|w_t) \big] + D_{\mathrm{KL}}\big( q_{\theta_t}(w_t) \| p_\psi(w_t) \big) \Big]. \tag{5}$$

However, the optimization in (5), as $N$ increases, becomes more and more expensive in terms of the memory footprint and the computational cost. We therefore wish to bypass this heavy optimization and to take advantage of the fact that individual KL terms indeed share the same structure. To this end, instead of introducing $N$ different variational distributions, we consider a parameterized family of distributions in the form of $q_{\phi(\cdot)}$, which is defined implicitly by a deep neural network $\phi$ taking as input either $d_t^l$ or $d_t^l$ plus $x_t$, that is, $q_{\phi(d_t^l)}$ or $q_{\phi(d_t^l, x_t)}$. Note that we cannot use entire $d_t$, since we do not have access to $y_t$ during meta-testing. This amortization technique was first introduced in the case of *variational autoencoders* (Kingma & Welling, 2013; Rezende et al., 2014), and has been extended to Bayesian inference in the case of *neural processes* (Garnelo et al., 2018).

Since $d_t^l$ and $x_t$ are disjoint, the inference scheme is *inductive* for a variational posterior $q_{\phi(d_t^l)}$. As an example, MAML (Finn et al., 2017) takes $q_{\phi(d_t^l)}$ as the Dirac delta distribution, where $\phi(d_t^l) = \theta_t^K$, is the $K$-th iterate of the stochastic gradient descent

$$\theta_t^{k+1} = \theta_t^k + \eta \, \nabla_\theta \mathbb{E}_{w_t \sim q_{\theta_t^k}} \Big[ \log p(d_t^l|w_t) \Big] \text{ with } \theta_t^0 = \phi, \text{ a learnable initialization.} \tag{6}$$

In this work, we consider a *transductive* inference scheme with variational posterior $q_{\phi(d_t^l, x_t)}$. The inference process is shown in Figure 1(a, in green arrows). Replacing each $q_{\theta_t}$ in (5) by $q_{\phi(d_t^l, x_t)}$, the optimization problem becomes

$$\min_{\psi,f} \min_{\phi} \frac{1}{N} \sum_{t=1}^{N} \Big[ \mathbb{E}_{w_t \sim q_{\phi(d_t^l, x_t)}} \big[ -\log p_f(d_t|w_t) \big] + D_{\mathrm{KL}}\big( q_{\phi(d_t^l, x_t)}(w_t) \| p_\psi(w_t) \big) \Big]. \tag{7}$$

In a nutshell, the meta-model to be optimized includes the feature network $f$, the hyper-parameter $\psi$ from the empirical Bayes formulation and the amortization network $\phi$ from the variational inference.

## 3 UNROLLING EXACT INFERENCE WITH SYNTHETIC GRADIENTS

It is however non-trivial to design a proper network architecture for $\phi(d_t^l, x_t)$, since $d_t^l$ and $x_t$ are both sets. The strategy adopted by neural processes (Garnelo et al., 2018) is to aggregate the information from all individual examples via an averaging function. However, as pointed out by Kim et al.

(2019), such a strategy tends to underfit $x_t$ because the aggregation does not necessarily attain the most relevant information for identifying the task-specific parameter. Extensions, such as attentive neural process (Kim et al., 2019) and set transformer (Lee et al., 2019a), may alleviate this issue but come at a price of $O(n^2)$ time complexity. We instead design $\phi(d_t^l, x_t)$ to mimic the exact inference $\arg\min_{\theta_t} D_{KL}(q_{\theta_t}(w_t) \| p_{\psi,f}(w_t|d_t))$ by parameterizing the optimization process with respect to $\theta_t$. More specifically, consider the gradient descent on $\theta_t$ with step size $\eta$:

$$\theta_t^{k+1} = \theta_t^k - \eta \, \nabla_{\theta_t} D_{KL}\Big(q_{\theta_t^k}(w) \, \| \, p_{\psi,f}(w \,|\, d_t)\Big). \tag{8}$$

We would like to unroll this optimization dynamics up to the $K$-th step such that $\theta_t^K = \phi(d_t^l, x_t)$ while make sure that $\theta_t^K$ is a good approximation to the optimum $\theta_t^\star$, which consists of parameterizing

(a) the **initialization** $\theta_t^0$ and (b) the **gradient** $\nabla_{\theta_t} D_{KL}(q_{\theta_t}(w_t) \, \| \, p_{\psi,f}(w_t|d_t))$.

By doing so, $\theta_t^K$ becomes a function of $\phi$, $\psi$ and $x_t$[2], we therefore realize $q_{\phi(d_t^l, x_t)}$ as $q_{\theta_t^K}$.

For (a), we opt to either let $\theta_t^0 = \lambda$ to be a global data-independent initialization as in MAML (Finn et al., 2017) or let $\theta_t^0 = \lambda(d_t^l)$ with a few supervisions from the support set, where $\lambda$ can be implemented by a permutation invariant network described in Gidaris & Komodakis (2018). In the second case, the features of the support set will be first averaged in terms of their labels and then scaled by a learnable vector of the same size.

For (b), the fundamental reason that we parameterize the gradient is because we do not have access to $y_t$ during the meta-testing phase, although we are able to follow (8) in meta-training to obtain $q_{\theta_t^\star}(w_t) \propto p_f(d_t|w_t)p_\psi(w_t)$. To make a consistent parameterization in both meta-training and meta-testing, we thus do not touch $y_t$ when constructing the variational posterior. Recall that the true gradient decomposes as

$$\nabla_{\theta_t} D_{KL}\Big(q_{\theta_t} \| p_{\psi,f}\Big) = \mathbb{E}_\epsilon\Big[\frac{1}{n} \sum_{i=1}^n \frac{\partial \ell_t(\hat{y}_{t,i}, y_{t,i})}{\partial \hat{y}_{t,i}} \frac{\partial \hat{y}_{t,i}}{\partial w_t} \frac{\partial w_t(\theta_t, \epsilon)}{\partial \theta_t}\Big] + \nabla_{\theta_t} D_{KL}\Big(q_{\theta_t} \| p_\psi\Big) \tag{9}$$

under a reparameterization $w_t = w_t(\theta_t, \epsilon)$ with $\epsilon \sim p(\epsilon)$, where all the terms can be computed without $y_t$ except for $\frac{\partial \ell_t}{\partial \hat{y}_{t,i}}$. Thus, we introduce a deep neural network $\xi(\hat{y}_{t,i})$ to synthesize it. The idea of synthetic gradients was originally proposed by Jaderberg et al. (2017) to parallelize the back-propagation. Here, the purpose of $\xi(\hat{y}_{,i})$ is to update $\theta_t$ regardless of the groundtruth labels, which is slightly different from its original purpose. Besides, we do not introduce an additional loss between $\xi(\hat{y}_{t,i})$ and $\frac{\partial \ell_t}{\partial \hat{y}_{t,i}}$ since $\xi(\hat{y}_{t,i})$ will be driven by the objective in (7). As an intermediate computation, the synthetic gradient is not necessarily a good approximation to the true gradient.

To sum up, we have derived a particular implementation of $\phi(d_t^l, x_t)$ by parameterizing the exact inference update, namely (8), without using the labels of the query set, where the meta-model $\phi$ includes an initialization network $\lambda$ and a synthetic gradient network $\xi$, such that $\phi(x_t) = \theta_t^K$, the $K$-th iterate of the following update:

$$\theta_t^{k+1} = \theta_t^k - \eta \Big[\mathbb{E}_\epsilon\Big[\frac{1}{n} \sum_{i=1}^n \xi(\hat{y}_{t,i}) \frac{\partial \hat{y}_{t,i}}{\partial w_t} \frac{\partial w_t(\theta_t^k, \epsilon)}{\partial \theta_t}\Big] + \nabla_{\theta_t} D_{KL}\Big(q_{\theta_t^k} \| p_\psi\Big)\Big]. \tag{10}$$

The overall algorithm is depicted in Algorithm 1. We also make a side-by-side comparison with MAML shown in Figure 1. Rather than viewing (10) as an optimization process, it may be more precise to think of it as a part of the computation graph created in the forward-propagation. The computation graph of the amortized inference is shown in Figure 2,

As an extension, if we were deciding to estimate the feature network $f$ in a Bayesian manner, we would have to compute higher-order gradients as in the case of MAML. This is inpractical from a computational point of view and needs technical simplifications (Nichol et al., 2018). By introducing a series of synthetic gradient networks in a way similar to Jaderberg et al. (2017), the computation will be decoupled into computations within each layer, and thus becomes more feasible. We see this as a potential advantage of our method and leave this to our future work[3].

---

[2]$\theta_t^K$ is also dependent of $f$. We deliberately remove this dependency to simplify the update of $f$.

[3]We do not insist on Bayesian estimation of the feature network because most Bayesian versions of CNNs underperform their deterministic counterparts.

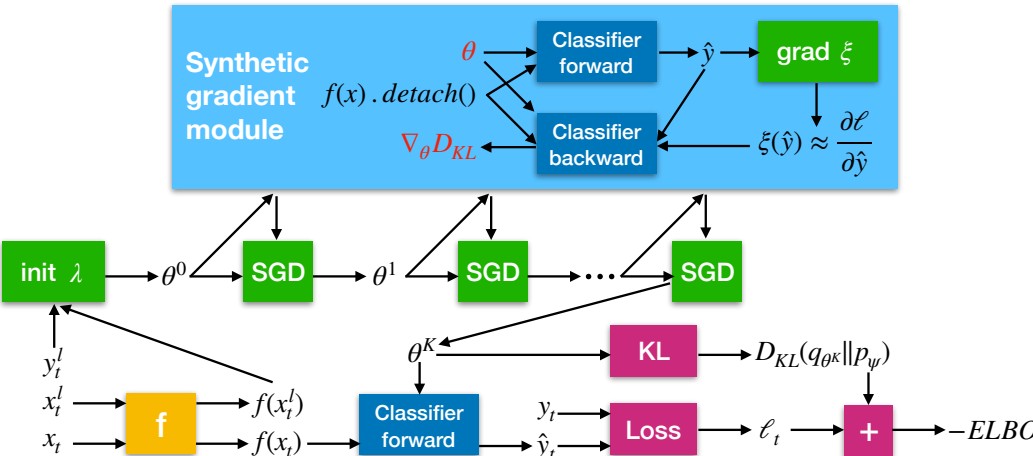

Figure 2: The computation graph to compute the negative ELBO, where the input and output of the synthetic gradient module are highlighted in red. The detach() is used to stop the back-propagation down to the feature network. Note that we do not include every computation for simplicity.

---

**Algorithm 1** Variational inference with synthetic gradients for empirical Bayes

1: **Input**: the dataset $\mathcal{D}$; the step size $\eta$; the number of inner iterations $K$; pretrained $f$.
2: Initialize the meta-models $\psi$, and $\phi = (\lambda, \xi)$.
3: **while** not converged **do**
4:     Sample a task $t$ and the associated query set $d_t$ (plus optionally the support set $d_t^l$).
5:     Compute the initialization $\theta_t^0 = \lambda$ or $\theta_t^0 = \lambda(d_t^l)$.
6:     **for** $k = 1, \ldots, K$ **do**
7:         Compute $\theta_t^k$ via (10).
8:     **end for**
9:     Compute $w_t = w_t(\theta_t^K, \epsilon)$ with $\epsilon \sim p(\epsilon)$.
10:    Update $\psi \leftarrow \psi - \eta \nabla_\psi D_{\text{KL}}(q_{\theta_t^K(\psi)} \| p_\psi)$.
11:    Update $\phi \leftarrow \phi - \eta \nabla_\phi D_{\text{KL}}(q_{\phi(x_t)} \| p_f \cdot p_\psi)$.
12:    Optionally, update $f \leftarrow f + \eta \nabla_f \log p_f(d_t | w_t)$.
13: **end while**

---

# 4   GENERALIZATION ANALYSIS OF EMPIRICAL BAYES VIA THE CONNECTION TO INFORMATION BOTTLENECK

The learning of empirical Bayes (EB) models follows the frequentist's approach, therefore, we can use frequentist's tool to analyze the model. In this section, we study the generalization ability of the empirical Bayes model through its connection to a variant of the information bottleneck principle Achille & Soatto (2017); Tishby et al. (2000).

**Abstract form of empirical Bayes**   From (3), we see that the empirical Bayes model implies a simpler joint distribution since

$$\log p_{\psi, f}(w_1, \ldots, w_N, \mathcal{D}) = \sum_{t=1}^{N} \log p_f(d_t | w_t) + \log p_\psi(w_t), \tag{11}$$

which is equal to the log-density of $N$ iid samples drawn from the joint distribution

$$p(w, d, t) \equiv p_{\psi, f}(w, d, t) = p_f(d | w, t) p_{\psi, f}(w) p(t)^4 \tag{12}$$

up to a constant if we introduce a random variable to represent the task and assume $p(t)$ is an uniform distribution. We thus see that this joint distribution embodies the *generative process* of empirical Bayes. Correspondingly, there is another graphical model of the joint distribution characterizes the

*inference process* of the empirical Bayes:

$$q(w, d, t) \equiv q_\phi(w, d, t) = q_\phi(w|d, t)q(d|t)q(t), \tag{13}$$

where $q_\phi(w|d, t)$ is the abstract form of the variational posterior with amortization, includes both the inductive form and the transductive form. The coupling between $p(w, d, t)$ and $q(w, d, t)$ is due to $p(t) \equiv q(t)$ as we only have access to tasks through sampling.

We are interested in the case that the number of tasks $N \to \infty$, such as the few-shot learning paradigm proposed by Vinyals et al. (2016), in which the objective of (7) can be rewritten in an abstract form of

$$\mathbb{E}_{q(t)}\mathbb{E}_{q(d|t)}\Big[\mathbb{E}_{q(w|d,t)}\big[-\log p(d|w, t)\big] + D_{\mathrm{KL}}\big(q(w|d, t)\|p(w)\big)\Big]. \tag{14}$$

In fact, optimizing this objective is the same as optimizing (7) from a stochastic gradient descent point of view.

The learning of empirical Bayes with amortized variational inference can be understood as a variational EM in the sense that the E-step amounts to aligning $q(w|d, t)$ with $p(w|d, t)$ while the M-step amounts to adjusting the likelihood $p(d|w, t)$ and the prior $p(w)$.

**Connection to information bottleneck**    The following theorem shows the connection between (14) and the information bottleneck principle.

**Theorem 1.** *Given distributions $q(w|d, t)$, $q(d|t)$, $q(t)$, $p(w)$ and $p(d|w, t)$, we have*

$$(14) \geq I_q(w; d|t) + H_q(d|w, t), \tag{19}$$

*where $I_q(w; d|t) := D_{KL}\big(q(w, d|t)\|q(w|t)q(d|t)\big)$ is the conditional mutual information and $H_q(w|d, t) := \mathbb{E}_{q(w,d,t)}[-\log q(w|d, t)]$ is the conditional entropy. The equality holds when*

$$\forall t \colon D_{KL}(q(w|t)\|p(w)) = 0 \ and \ D_{KL}(q(d|w, t)\|p(d|w, t)) = 0.$$

In fact, the lower bound on (14) is an extention of the information bottleneck principle (Achille & Soatto, 2017) under the multi-task setting, which, together with the synthetic gradient based variational posterior, inspire the name **synthetic information bottleneck** of our method. The tightness of the lower bound depends on both the parameterizations of $p_f(d|w, t)$ and $p_\psi(w)$ as well as the optimization of (14). It thus can be understood as how well we can align the inference process with the generative process. From an inference process point of view, for a given $q(w|d, t)$, the optimal likelihood and prior have been determined. In theory, we only need to find the optimal $q(w|d, t)$ using the information bottleneck in (19). However, in practice, minimizing (14) is more straightforward.

**Generalization of empirical Bayes meta-learning**    The connection to information bottleneck suggests that we can eliminate $p(d|w, t)$ and $p(w)$ from the generalization analysis of empirical Bayes meta-learning and define the generalization error by $q(w, d, t)$ only. To this end, we first identify the *empirical risk* for a single task $t$ with respect to particular weights $w$ and dataset $d$ as

$$L_t(w, d) := \frac{1}{n}\sum_{i=1}^{n}\ell_t(\hat{y}_i(f(x_i), w), y_i). \tag{15}$$

The *true risk* for task $t$ with respect to $w$ is then the expected empirical risk $\mathbb{E}_{d \sim q(d|t)}L_t(w, d)$. Now, we define the *generalization error* with respect to $q(w, d, t)$ as the average of the difference between the true risk and the empirical risk over all possible $t, d, w$:

$$\begin{aligned}
\mathrm{gen}(q) &:= \mathbb{E}_{q(t)q(d|t)q(w|d,t)}\Big[\mathbb{E}_{d \sim q(d|t)}L_t(w, d) - L_t(w, d)\Big] \\
&= \mathbb{E}_{q(t)q(d|t)q(w|t)}L_t(w, d) - \mathbb{E}_{q(t)q(d|t)q(w|d,t)}L_t(w, d), \tag{16}
\end{aligned}$$

where $q(w|t)$ is the *aggregated posterior* of task $t$.

Next, we extend the result from Xu & Raginsky (2017) and derive a data-dependent upper bound for $\mathrm{gen}(q)$ using mutual information.

**Theorem 2.** *Denote by $z = (x, y)$. If $\ell_t(\hat{y}_i(f(x_i), w), y_i) \equiv \ell_t(w, z_i)$ is $\sigma$-subgaussian under $q(w|t)q(z|t)$, then $L_t(w, d)$ is $\sigma/\sqrt{n}$-subgaussian under $q(w|t)q(d|t)$ due to the iid assumption, and*

$$\big|gen(q)\big| \leq \sqrt{\frac{2\sigma^2}{n}I_q(w; d|t)}. \tag{30}$$

Plugging this back to Theorem 1, we obtain a different interpretation for the empirical Bayes ELBO.

**Corollary 1.** *If $\ell_t$ is chosen to be the negative log-likelihood, minimizing the population objective of empirical Bayes meta-learning amounts to minimizing both the expected generalization error and the expected empirical risk:*

$$(14) \geq \frac{n}{2\sigma^2} gen(q)^2 + \mathbb{E}_{q(t)q(d|t)q(w|d,t)} L_t(w, d). \tag{17}$$

The Corollary 1 suggests that (14) amounts to minimizing a regularized empirical risk minimization. In general, there is a tradeoff between the generalization error and the empirical risk controlled by the coefficient $\frac{n}{2\sigma^2}$, where $n = |d|$ is the cardinality of $d$. If $n$ is small, then we are in the overfitting regime. This is the case of the inductive inference with variational posterior $q(w|d^l, t)$, where the support set $d^l$ is fairly small by the definition of few-shot learning. On the other hand, if we were following the transductive setting, we expect to achieve a small generalization error since the implemented variational posterior is a better approximation to $q(w|d, t)$. However, keeping increasing $n$ will potentially over-regularize the model and thus yield negative effect. An empirical study on varying $n$ can be found in Table 5 in Appendix D.

## 5 EXPERIMENTS

In this section, we first validate our method on few-shot learning, and then on zero-shot learning (no support set and no class description are available). Note that many meta-learning methods cannot do zero-shot learning since they rely on the support set.

### 5.1 FEW-SHOT CLASSIFICATION

We compare SIB with state-of-the-art methods on few-shot classification problems. Our code is available at `https://github.com/amzn/xfer`.

#### 5.1.1 SETUP

**Datasets** We choose standard benchmarks of few-shot classification for this experiment. Each benchmark is composed of disjoint training, validation and testing classes. **MiniImageNet** is proposed by Vinyals et al. (2016), which contains 100 classes, split into 64 training classes, 16 validation classes and 20 testing classes; each image is of size 84×84. **CIFAR-FS** is proposed by Bertinetto et al. (2018), which is created by dividing the original CIFAR-100 into 64 training classes, 16 validation classes and 20 testing classes; each image is of size 32×32.

**Evaluation metrics** In few-shot classification, a task (aka episode) $t$ consists of a *query set* $d_t$ and a *support set* $d_t^l$. When we say the task $t$ is *k-way-$n^l$-shot* we mean that $d_t^l$ is formed by first sampling $k$ classes from a pool of classes; then, for each sampled class, $n^l$ examples are drawn and a new label taken from $\{0, \ldots, k-1\}$ is assigned to these examples. By default, each query set contains $15k$ examples. The goal of this problem is to predict the labels of the query set, which are provided as ground truth during training. The evaluation is the average classification accuracy over tasks.

**Network architectures** Following Gidaris & Komodakis (2018); Qiao et al. (2018); Gidaris et al. (2019), we implement $f$ by a 4-layer convolutional network (Conv-4-64 or Conv-4-128[5]) or a WideResNet (WRN-28-10) (Zagoruyko & Komodakis, 2016). We pretrain the feature network $f(\cdot)$ on the 64 training classes for a stardard 64-way classification. We reuse the feature averaging network proposed by Gidaris & Komodakis (2018) as our initialization network $\lambda(\cdot)$, which basically averages the feature vectors of all data points from the same class and then scales each feature dimension differently by a learned coefficient. For the synthetic gradient network $\xi(\cdot)$, we implement a three-layer MLP with hidden-layer size $8k$. Finally, for the predictor $\hat{y}_{ij}(\cdot, w_i)$, we adopt the cosine-similarity based classifier advocated by Chen et al. (2019) and Gidaris & Komodakis (2018).

---

[5]Conv-4-64 consists of 4 convolutional blocks each implemented with a $3 \times 3$ convolutional layer followed by BatchNorm + ReLU + $2 \times 2$ max-pooling units. All blocks of Conv-4-64 have 64 feature channels. Conv-4-128 has 64 feature channels in the first two blocks and 128 feature channels in the last two blocks.

| Method | Backbone | MiniImageNet, 5-way | | CIFAR-FS, 5-way | |
|---|---|---|---|---|---|
| | | 1-shot | 5-shot | 1-shot | 5-shot |
| Matching Net (Vinyals et al., 2016) | Conv-4-64 | 44.2% | 57% | – | – |
| MAML (Finn et al., 2017) | Conv-4-64 | 48.7±1.8% | 63.1±0.9% | 58.9±1.9% | 71.5±1.0% |
| Prototypical Net (Snell et al., 2017) | Conv-4-64 | 49.4±0.8% | 68.2±0.7% | 55.5±0.7% | 72.0±0.6% |
| Relation Net (Sung et al., 2018) | Conv-4-64 | 50.4±0.8% | 65.3±0.7% | 55.0±1.0% | 69.3±0.8% |
| GNN (Satorras & Bruna, 2017) | Conv-4-64 | 50.3% | 66.4% | 61.9% | 75.3% |
| R2-D2 (Bertinetto et al., 2018) | Conv-4-64 | 49.5±0.2% | 65.4±0.2% | 62.3±0.2% | 77.4±0.2% |
| TPN (Liu et al., 2018) | Conv-4-64 | 55.5% | 69.9% | – | – |
| Gidaris et al. (2019) | Conv-4-64 | 54.8±0.4% | **71.9±0.3%** | 63.5±0.3% | **79.8±0.2%** |
| SIB $K$=0 (*Pre-trained feature*) | Conv-4-64 | 50.0±0.4% | 67.0±0.4% | 59.2±0.5% | 75.4±0.4% |
| SIB $\eta$=1e-3, $K$=3 | Conv-4-64 | **58.0±0.6%** | 70.7±0.4% | **68.7±0.6%** | 77.1±0.4% |
| SIB $\eta$=1e-3, $K$=0 | Conv-4-128 | 53.62 ± 0.79% | 71.48 ± 0.64% | – | – |
| SIB $\eta$=1e-3, $K$=1 | Conv-4-128 | 58.74 ± 0.89% | 74.12 ± 0.63% | – | – |
| SIB $\eta$=1e-3, $K$=3 | Conv-4-128 | 62.59 ± 1.02% | 75.43 ± 0.67% | – | – |
| SIB $\eta$=1e-3, $K$=5 | Conv-4-128 | **63.26 ± 1.07%** | **75.73 ± 0.71%** | – | – |
| TADAM (Oreshkin et al., 2018) | ResNet-12 | 58.5±0.3% | 76.7±0.3% | – | – |
| SNAIL (Santoro et al., 2017) | ResNet-12 | 55.7±1.0% | 68.9±0.9% | – | – |
| MetaOptNet-RR (Lee et al., 2019b) | ResNet-12 | 61.4±0.6% | 77.9±0.5% | 72.6±0.7% | 84.3±0.5% |
| MetaOptNet-SVM | ResNet-12 | 62.6±0.6% | 78.6±0.5% | 72.0±0.7% | 84.2±0.5% |
| CTM (Li et al., 2019) | ResNet-18 | 64.1±0.8% | **80.5±0.1%** | – | – |
| Qiao et al. (2018) | WRN-28-10 | 59.6±0.4% | 73.7±0.2% | – | – |
| LEO (Rusu et al., 2019) | WRN-28-10 | 61.8±0.1% | 77.6±0.1% | – | – |
| Gidaris et al. (2019) | WRN-28-10 | 62.9±0.5% | 79.9±0.3% | 73.6±0.3% | **86.1±0.2%** |
| SIB $K$=0 (*Pre-trained feature*) | WRN-28-10 | 60.6±0.4% | 77.5±0.3% | 70.0±0.5% | 83.5±0.4% |
| SIB $\eta$=1e-3, $K$=1 | WRN-28-10 | 67.3±0.5% | 78.8±0.4% | 76.8±0.5% | 84.9±0.4% |
| SIB $\eta$=1e-3, $K$=3 | WRN-28-10 | 69.6±0.6 % | 78.9±0.4% | 78.4±0.6% | 85.3±0.4% |
| SIB $\eta$=1e-3, $K$=5 | WRN-28-10 | **70.0±0.6%** | 79.2±0.4% | **80.0±0.6%** | 85.3±0.4% |

Table 2: Average classification accuracies (with 95% confidence intervals) on the test-set of Mini-ImageNet and CIFAR-FS. For evaluation, we sample 2000 and 5000 episodes respectively for MiniImageNet and CIFAR-FS and test three different architectures as the feature extractor: Conv-4-64, Conv-4-128 and WRN-28-10. We train SIB with learning rate 0.001 and try different numbers of synthetic gradient steps $K$.

**Training details** We run SGD with batch size 8 for 40000 steps, where the learning rate is fixed to $10^{-3}$. During training, we freeze the feature network. To select the best hyper-parameters, we sample 1000 tasks from the validation classes and reuse them at each training epoch.

### 5.1.2 COMPARISON TO STATE-OF-THE-ART META-LEARNING METHODS

In Table 2, we show a comparison between the state-of-the-art approaches and several variants of our method (varying $K$ or $f(\cdot)$). Apart from SIB, TPN (Liu et al., 2018) and CTM (Li et al., 2019) are also transductive methods.

First of all, comparing SIB ($K = 3$) to SIB ($K = 0$), we observe a clear improvement, which suggests that, by taking a few synthetic gradient steps, we do obtain a better variational posterior as promised. For 1-shot learning, SIB (when $K = 3$ or $K = 5$) significantly outperforms previous methods on both MiniImageNet and CIFAR-FS. For 5-shot learning, the results are also comparable. It should be noted that the performance boost is consistently observed with different feature networks, which suggests that SIB is a general method for few-shot learning.

However, we also observe a potential limitation of SIB: when the support set is relatively large, e.g., 5-shot, with a good feature network like WRN-28-10, the initialization $\theta_t^0$ may already be close to some local minimum, making the updates later less important.

For 5-shot learning, SIB is sligtly worse than CTM (Li et al., 2019) and/or Gidaris et al. (2019). CMT (Li et al., 2019) can be seen as an alternative way to incorporate transduction – it measures the similarity between a query example and the support set while making use of intra- and inter-class relationships. Gidaris et al. (2019) uses in addition the self-supervision as an auxilary loss to learn a richer and more transferable feature model. Both ideas are complementary to SIB. We leave these extensions to our future work.

## 5.2 ZERO-SHOT REGRESSION: SPINNING LINES

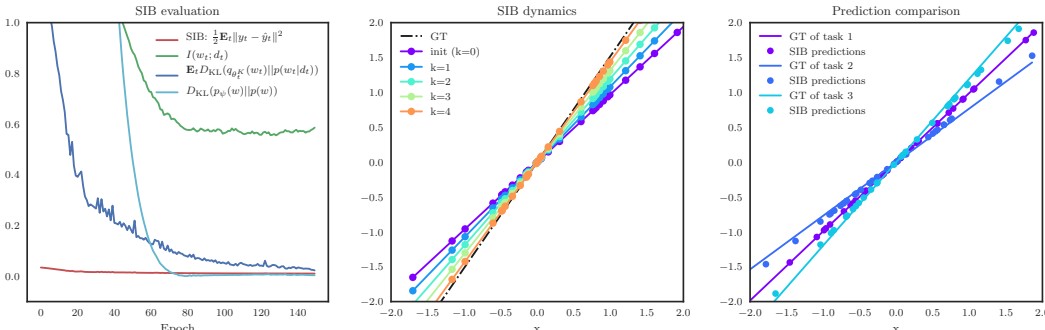

**Figure 3: Left**: the mean-square errors on $D_{\text{test}}$, $\mathbb{E}_t D_{\text{KL}}(q_{\theta_t^K}(w_t)\|p(w_t|d_t))$, $D_{\text{KL}}(p_\psi(w)\|p(w))$ and the estimate of $I(w;d) \approx \mathbb{E}_t D_{\text{KL}}(q_{\theta_t^K}(w_t)\|p_\psi(w_t))$. **Middle**: the predicted $y$'s by $y = \theta_t^k x$ for $k = 0, \ldots, 4$. **Right**: the predictions of SIB.

Since our variational posterior relies only on $x_t$, SIB is also applicable to zero-shot problems (i.e., no support set available). We first look at a toy multi-task problem, where $I(w_t; d_t)$ is tractable.

Denote by $D_{\text{train}} := \{d_t\}_{t=1}^N$ the train set, which consists of datasets of size $n$: $d = \{(x_i, y_i)\}_{i=1}^n$. We construct a dataset $d$ by firstly sampling iid Gaussian random variables as inputs: $x_i \sim \mathcal{N}(\mu, \sigma^2)$. Then, we generate the weight for each dataset by calculating the mean of the inputs and shifting with a Gaussian random variable $\epsilon_w$: $w = \frac{1}{n}\sum_i x_i + \epsilon_w$, $\epsilon_w \sim \mathcal{N}(\mu_w, \sigma_w^2)$. The output for $x_i$ is $y_i = w \cdot x_i$. We decide ahead of time the hyperparameters $\mu, \sigma, \mu_w, \sigma_w$ for generating $x_i$ and $y_i$. Recall that a weighted sum of iid Gaussian random variables is still a Gaussian random variable. Specifically, if $w = \sum_i c_i x_i$ and $x_i \sim \mathcal{N}(\mu_i, \sigma_i^2)$, then $w \sim \mathcal{N}(\sum_i c_i \mu_i, \sum_i c_i^2 \sigma_i^2)$. Therefore, we have $p(w) = \mathcal{N}(\mu + \mu_w, \frac{1}{n}\sigma^2 + \sigma_w^2)$. On the other hand, if we are given a dataset $d$ of size $n$, the only uncertainty about $w$ comes from $\epsilon_w$, that is, we should consider $x_i$ as a constant given $d$. Therefore, the posterior $p(w|d) = \mathcal{N}(\frac{1}{n}\sum_{i=1}^n x_i + \mu_w, \sigma_w^2)$. We use a simple implementation for SIB: The variational posterior is realized by

$$q_{\theta_t^K}(w) = \mathcal{N}(\theta_t^K, \sigma_w), \ \theta_t^{k+1} = \theta_t^k - 10^{-3}\sum_{i=1}^n x_i \xi(\theta_t^k x_i), \ \text{and} \ \theta_t^0 = \lambda \in \mathbb{R}; \qquad (18)$$

$\ell_t$ is a mean squared error, implies that $p(y|x, w) = \mathcal{N}(wx, 1)$; $p_\psi(w)$ is a Gaussian distribution with parameters $\psi \in \mathbb{R}^2$; The synthetic gradient network $\xi$ is a three-layer MLP with hidden size 8.

In the experiment, we sample 240 tasks respectively for both $D_{\text{train}}$ and $D_{\text{test}}$. We learn SIB and BNN on $D_{\text{train}}$ for 150 epochs using the ADAM optimizer (Kingma & Ba, 2014), with learning rate $10^{-3}$ and batch size 8. Other hyperparameters are specified as follows: $n = 32, K = 3, \mu = 0, \sigma = 1, \mu_w = 1, \sigma_w = 0.1$. The results are shown in Figure 3. On the left, both $D_{\text{KL}}(q_{\theta_t^K}(w_t)\|p(w_t|d_t))$ and $D_{\text{KL}}(p_\psi(w)\|p(w))$ are close to zero indicating the success of the learning. More interestingly, in the middle, we see that $\theta_t^0, \theta_t^1, \ldots, \theta_t^4$ evolves gradually towards the ground truth, which suggests that the synthetic gradient network is able to identify the descent direction after meta-learning.

## 6 CONCLUSION

We have presented an empirical Bayesian framework for meta-learning. To enable an efficient variational inference, we followed the amortized inference paradigm, and proposed to use a transductive scheme for constructing the variational posterior. To implement the transductive inference, we make use of two neural networks: a synthetic gradient network and an initialization network, which together enables a synthetic gradient descent on the unlabeled data to generate the parameters of the amortized variational posterior dynamically. We have studied the theoretical properties of the proposed framework and shown that it yields performance boost on MiniImageNet and CIFAR-FS for few-shot classification.

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

## APPENDIX

### A  PROOFS

**Theorem 1.** *Given distributions $q(w|d,t)$, $q(d|t)$, $q(t)$, $p(w)$ and $p(d|w,t)$, we have*

$$(14) \geq I_q(w;d|t) + H_q(d|w,t), \tag{19}$$

*where $I_q(w;d|t) := D_{KL}\big(q(w,d|t)\|q(w|t)q(d|t)\big)$ is the conditional mutual information and $H_q(w|d,t) := \mathbb{E}_{q(w,d,t)}[-\log q(w|d,t)]$ is the conditional entropy. The equality holds when*

$$\forall t: D_{KL}(q(w|t)\|p(w)) = 0 \text{ and } D_{KL}(q(d|w,t)\|p(d|w,t)) = 0.$$

*Proof.* Denote by $q(w|t) := \mathbb{E}_{q(d|t)}q(w|d,t)q(d|t)$ the aggregated posterior of task $t$. (14) can be decomposed as

$$\mathbb{E}_{q(t)}\mathbb{E}_{q(d|t)}\Big[\mathbb{E}_{q(w|d,t)}\big[-\log p(d|w,t)\big] + D_{\mathrm{KL}}\big(q(w|d,t)\|p(w)\big)\Big] \tag{20}$$

$$= \mathbb{E}_{q(t)}\mathbb{E}_{q(d|t)}\mathbb{E}_{q(w|d,t)}\Big[\log \frac{q(w|d,t)q(w|t)}{p(d|w,t)p(w)q(w|t)}\Big] \tag{21}$$

$$= \mathbb{E}_{q(t)}\mathbb{E}_{q(d|t)}\mathbb{E}_{q(w|d,t)}\Big[\log \frac{q(w|d,t)}{q(w|t)}\Big] + \mathbb{E}_{q(t)}\mathbb{E}_{q(d|t)}\mathbb{E}_{q(w|d,t)}\Big[-\log p(d|w,t)\Big]$$

$$\quad + \mathbb{E}_{q(t)}\mathbb{E}_{q(d|t)}\mathbb{E}_{q(w|d,t)}\Big[\log \frac{q(w|t)}{p(w)}\Big] \tag{22}$$

$$= I_q(w;d|t) + H_{q,p}(d|w,t) + \mathbb{E}_{q(t)}D_{\mathrm{KL}}(q(w|t)\|p(w)) \tag{23}$$

$$\geq I_q(w;d|t) + H_{q,p}(d|w,t). \tag{24}$$

The inequality is because $D_{\mathrm{KL}}(q(w|t)\|p(w)) \geq 0$ for all $t$'s. Besides, we used the notation $H_{q,p}$, which is the conditional cross entropy. Recall that $D_{\mathrm{KL}}\big(q(d|w,t)\|p(d|w,t)\big) = -H_q(d|w,t) + H_{q,p}(d|w,t) \geq 0$. We attain the lower bound as desired if this inequality is applied to replace $H_{q,p}(d|w,t)$ by $H_q(d|w,t)$. $\qquad\square$

The following lemma and theorem show the connection between $I_q(w;d|t)$ and the generalization error. We first extend Xu (2016, Lemma 4.2).

**Lemma 1.** *If, for all $t$, $f_t(X,Y)$ is $\sigma$-subgaussain under $P_X \otimes P_Y$, then*

$$\Big|\mathbb{E}_{P(T)}\Big[\mathbb{E}_{P(X,Y|T)}f_T(X,Y) - \mathbb{E}_{P(X|T)P(Y|T)}f_T(X,Y)\Big]\Big| \leq \sqrt{2\sigma^2 I(X;Y|T)}. \tag{25}$$

*Proof.* The proof is adapted from the proof of Xu (2016, Lemma 4.2).

$$LHS \leq \mathbb{E}_{P(T)}\Big|\mathbb{E}_{P(X,Y|T)}f_T(X,Y) - \mathbb{E}_{P(X|T)P(Y|T)}f_T(X,Y)\Big| \tag{26}$$

$$\leq \mathbb{E}_{P(T)}\sqrt{2\sigma^2 D_{\mathrm{KL}}(P(X,Y|T)\|P(X|T)P(Y|T))} \tag{27}$$

$$\leq \sqrt{2\sigma^2 \mathbb{E}_{P(T)}D_{\mathrm{KL}}(P(X,Y|T)\|P(X|T)P(Y|T))} \tag{28}$$

$$= \sqrt{2\sigma^2 I(X;Y|T)}. \tag{29}$$

The second inequality was due to the Donsker-Varadhan variational representation of KL divergence and the definition of subgaussain random variable. $\qquad\square$

**Theorem 2.** *Denote by $z = (x,y)$. If $\ell_t(\hat{y}_i(f(x_i),w),y_i) \equiv \ell_t(w,z_i)$ is $\sigma$-subgaussian under $q(w|t)q(z|t)$, then $L_t(w,d)$ is $\sigma/\sqrt{n}$-subgaussian under $q(w|t)q(d|t)$ due to the iid assumption, and*

$$\big|gen(q)\big| \leq \sqrt{\frac{2\sigma^2}{n}I_q(w;d|t)}. \tag{30}$$

*Proof.* First, if $\ell_t(\hat{y}(f(x),w),y)$ is $\sigma$-subgaussian under $q(w|t)q(z|t)$, by definition,

$$\mathbb{E}_{q(w|t)q(z|t)}\exp(\lambda\ell_t(w,z)) \leq \exp(\lambda\mathbb{E}_{q(w|t)q(z|t)}\ell_t(w,z))\exp(\lambda^2\sigma^2/2) \tag{31}$$

It is straightforward to show $L_t(w,d)$ is $\sigma/\sqrt{n}$-subgaussian since

$$\mathbb{E}_{q(w|t)q(d|t)}\exp(\lambda L_t(w,d)) = \prod_{i=1}^{n}\mathbb{E}_{w,z_i}\exp(\frac{\lambda}{n}\ell_t(w,z_i)) \tag{32}$$

$$\leq \prod_{i=1}^{n}\exp\Big(\frac{\lambda}{n}\mathbb{E}_{w,z_i}\ell_t(w,z_i) + \frac{\lambda^2\sigma^2}{2n^2}\Big) \tag{33}$$

$$= \exp\Big(\lambda\mathbb{E}_{w,z}\ell_t(w,z)\Big)\exp(\frac{\lambda^2\sigma^2}{2n}) \tag{34}$$

$$= \exp\Big(\lambda\mathbb{E}_{q(w|t)q(d|t)}L_t(w,d)\Big)\exp(\frac{\lambda^2(\sigma/\sqrt{n})^2}{2}). \tag{35}$$

| Method | Art | Cartoon | Sketch | Photo | Average |
|---|---|---|---|---|---|
| JiGen (Carlucci et al., 2019) | 84.9% | 81.1% | 79.1% | 98.0% | 85.7% |
| Rot (Xu et al., 2019) | 88.7% | 86.4% | 74.9% | 98.0% | 87.0% |
| SIB-Rot $K = 0$ | 85.7% | 86.6% | 80.3% | 98.3% | 87.7% |
| SIB-Rot $K = 3$ | **88.9%** | **89.0%** | **82.2%** | **98.3%** | **89.6%** |

Table 3: Multi-source domain adaptation results on PACS with ResNet-18 features. Three domains are used as the source domains keeping the fourth one as target.

By Lemma 1, we have

$$\left|\text{gen}(q)\right| = \left|\mathbb{E}_{q(t)}\Big[\mathbb{E}_{q(w|d,t)q(d|t)}L_t(w,d) - \mathbb{E}_{q(w|t)q(d|t)}L_t(w,d)\Big]\right| \tag{36}$$

$$\leq \sqrt{\frac{2\sigma^2}{n}I(w;d|t)} \tag{37}$$

as desired. □

## B    ZERO-SHOT CLASSIFICATION: UNSUPERVISED MULTI-SOURCE DOMAIN ADAPTATION

A more interesting zero-shot multi-task problem is unsupervised domain adaptation. We consider the case where there exists multiple source domains and a unlabeled target domain. In this case, we treat each minibatch as a task. This makes sense because the difference in statistics between two minibatches are much larger than in the traditional supervised learning. The experimental setup is similar to few-shot classification described in Section 5.1, except that we do not have a support set and the class labels between two tasks are the same. Hence, it is possible to explore the relationship between class labels and *self-supervised* labels to implement the initialization network $\lambda$ without resorting to support set. We reuse the same model implementation for SIB as described in Section 5.1. The only difference is the initialization network. Denote by $z_t := \{z_{t,i}\}_{i=1}^n$ the set of self-supervised labels of task $t$, the initialization network $\lambda$ is implemented as follows:

$$\theta_t^0 = \lambda - \eta \nabla_\theta L_t\Big(\hat{z}_t\big(\hat{y}_t(f(x_t), w_t(\theta, \epsilon)), f(x_t)\big), z_t\Big), \tag{38}$$

where $\lambda$[6] is a global initialization similar to the one used by MAML; $L_t$ is the self-supervised loss, $\hat{z}_t$ is the set of predictions of the self-supervised labels. One may argue that $\theta_t^0 = \lambda$ would be a simpler solution. However, it is insufficient since the gap between two domains can be very large. The initial solution yielded by (38) is more dynamic in the sense that $\theta_t^0$ is adapted taking into account the information from $x_t$.

In terms of experiments, we test SIB on the PACS dataset (Li et al., 2017a), which has 7 object categories and 4 domains (Photo, Art Paintings, Cartoon and Sketches), and compare with state-of-the-art algorithms for unsupervised domain adaptation. We follow the standard experimental setting (Carlucci et al., 2019), where the feature network is implemented by ResNet-18. We assign a self-supervised label $z_{t,i}$ to image $i$ by rotating the image by a predicted degree. This idea was originally proposed by Gidaris et al. (2018) for representation learning and adopted by Xu et al. (2019) for domain adaptation. The training is done by running ADAM for 60 epochs with learning rate $10^{-4}$. We take each domain in turns as the target domain. The results are shown in Table 3. It can be seen that SIB-Rot ($K = 3$) improves upon the baseline SIB-Rot ($K = 0$) for zero-shot classification, which also outperforms state-of-the-art methods when the baseline is comparable.

## C    IMPORTANCE OF SYNTHETIC GRADIENTS

To further verify the effectiveness of the synthetic gradient descent, we implement an inductive version of SIB inspired by MAML, where the initialization $\theta_t^0$ is generated exactly the same way as SIB using $\lambda(d_t^l)$, but it then follows the iterations in (6) as in MAML rather than follows the iterations in (10) as in standard SIB.

---

[6] $\lambda$ is overloaded to be both the network and its parameters.

We conduct an experiment on CIFAR-FS using Conv-4-64 feature network. The results are shown in Table 4. It can be seen that there is no improvement over SIB ($K = 0$) suggesting that the inductive approach is insufficient.

| | | inductive SIB | | SIB | | | |
| | | Training on 1-shot Testing on | | Training on 1-shot Testing on | | Training on 5-shot Testing on | |
| $K$ | $\eta$ | 1-shot | 5-shot | 1-shot | 5-shot | 1-shot | 5-shot |
|---|---|---|---|---|---|---|---|
| 0 | - | 59.7±0.5% | 75.5±0.4% | 59.2±0.5% | 75.4±0.4% | 59.2±0.5% | 75.4±0.4% |
| 1 | 1e-1 | 59.8±0.5% | 71.2±0.4% | 65.3±0.6% | 75.8±0.4% | 64.5±0.6% | 77.3±0.4% |
| 3 | 1e-1 | 59.6±0.5% | 75.9±0.4% | 65.0±0.6% | 75.0±0.4% | 64.0±0.6% | 77.0±0.4% |
| 5 | 1e-1 | 59.9±0.5% | 74.9±0.4% | 66.0±0.6% | 76.3±0.4% | 64.0±0.5% | 76.8±0.4% |
| 1 | 1e-2 | 59.7±0.5% | 75.5±0.4% | 67.8±0.6% | 74.3±0.4% | 63.6±0.6% | 77.3±0.4% |
| 3 | 1e-2 | 59.5±0.5% | 75.8±0.4% | 68.6±0.6% | 77.4±0.4% | 67.8±0.6% | 78.5±0.4% |
| 5 | 1e-2 | 59.8±0.5% | 75.7±0.4% | 67.4±0.6% | 72.6±0.6% | 67.7±0.7% | 77.7±0.4% |
| 1 | 1e-3 | 59.5±0.5% | 75.6±0.4% | 66.2±0.6% | 75.7±0.4% | 64.6±0.6% | 78.1±0.4% |
| 3 | 1e-3 | 59.9±0.5% | 75.9±0.4% | 68.7±0.6% | 77.1±0.4% | 66.8±0.6% | 78.4±0.4% |
| 5 | 1e-3 | 59.4±0.5% | 75.7±0.4% | 69.1±0.6% | 76.7±0.4% | 66.7±0.6% | 78.5±0.4% |
| 1 | 1e-4 | 58.8±0.5% | 75.5±0.4% | 59.0±0.5% | 75.7±0.4% | 59.3±0.5% | 75.7±0.4% |
| 3 | 1e-4 | 59.4±0.5% | 75.9±0.4% | 58.9±0.5% | 75.6±0.4% | 59.3±0.5% | 75.9±0.4% |
| 5 | 1e-4 | 59.3±0.5% | 75.3±0.4% | 60.1±0.5% | 76.0±0.4% | 60.5±0.5% | 76.4±0.4% |

Table 4: Average 5-way classification accuracies (with 95% confidence intervals) with Conv-4-64 on the test set of CIFAR-FS. For each test, we sample 5000 episodes containing 5 categories (5-way) and 15 queries in each category. We report the results with using different learning rate $\eta$ as well as different number of updates $K$. Note that $K = 0$ is the performance only using the pre-trained feature.

## D  VARYING THE SIZE OF THE QUERY SET

We notice that changing the size of $d_t$ (i.e., $n$) during training does make a difference on testing. The results are shown in Table 5.

| $n$ | 5-way, 5-shot | | 5-way, 1-shot | |
| | Validation | Test | Validation | Test |
|---|---|---|---|---|
| 3 | 77.97 ± 0.34% | 75.91 ± 0.66% | 63.60 ± 0.52% | 61.32 ± 1.02% |
| 5 | 78.14 ± 0.35% | 76.01 ± 0.66% | 64.67 ± 0.55% | 62.50 ± 1.02% |
| 10 | **78.30 ± 0.35%** | **76.22 ± 0.66%** | **65.34 ± 0.56%** | **63.22 ± 1.04%** |
| 15 | 77.53 ± 0.35% | 75.43 ± 0.67% | 65.14 ± 0.55% | 62.59 ± 1.02% |
| 30 | 76.21 ± 0.35% | 74.04 ± 0.67% | 63.37 ± 0.53% | 60.96 ± 0.98% |
| 45 | 75.65 ± 0.36% | 73.27 ± 0.66% | 62.08 ± 0.51% | 59.59 ± 0.93% |

Table 5: Average classification accuracies on the validation set and the test set of Mini-ImageNet with backbone Conv-4-128. We modify the number of query images, i.e., $n$, for each episode to study the effect on generalization.

