# OpenReview forum: "Empirical Bayes Transductive Meta-Learning with Synthetic Gradients"
_ICLR.cc/2020/Conference — Accept (Poster)_

### Official Review · AnonReviewer2 · 2019-10-23
**Official Blind Review #2**

**Rating:** 6

**Review:**

The authors argue for the importance of transduction in few-shot learning approaches, and augment the empirical Bayes objective established in previous work (estimating the hyperpior $\psi$ in a frequentist fashion) so as to take advantage of the unlabelled test data.
Since the label is, by definition, unknown, a synthetic gradient is instead learned to parameterize the gradient of the variational posterior and tailor it for the transductive setting. The authors provide an analysis of the generalization ability of EB and term their method _synthetic information bottleneck_ (SIB) owing to parallels between its objective of that of the information bottleneck. SIB is tested on two standard few-shot image benchmarks in CIFAR-FS and MiniImageNet, exhibiting impressive performance and outperforming, in some cases by some margin, the baseline methods, in the 1- and 5-shot settings alike, in addition to a synthetic dataset.

The paper is technically sound and, for the most part, well-written, with the authors' motivations and explanation of the method conceived quite straightforwardly. The basic premise of using an estimated gradient to fashion an inductive few-shot learning algorithm into a transductive one is a natural and evidently potent one. The paper does, however, at times feel to be disjointed and, to an extent, lacking in focus. The respective advantages of EB and the repurposing of synthetic gradients to enable the transductive approach are clear to me, yet while they might indeed be obviously complementary, what is not obvious is the necessity of the pairing: it seems there is nothing prohibiting the substitution of the gradient for a learned surrogated just as well under a deterministic meta-initialization framework. As such, despite sporting impressive results on the image datasets, I am not convinced about how truly novel the method is when viewed as a whole.

On a similar note, while the theoretical analysis provided in section 4 was not unappreciated, and indeed it was interesting to see such a connection between EB with information theory rigorously made, it does feel a little out of place within the main text, especially since it is not specific to the transductive setting considered, nor even to the meta-learning setting more broadly. Rather, more experiments, per Appendix C, highlighting the importance of transduction and therein the synthetic gradients and its formulation would be welcome. Indeed, it is stated that an additional loss for training the synthetic gradient network to mimic the true gradient is unnecessary; while I agree with this conclusion, I likewise do not think it would hurt to explore use of the more explicit formulation.

Considering the authors argue specifically for the importance of transduction in the zero-shot learning regime, I think it would be reasonable to expect experiments substantiating this, and the strength of their method in this regard, on non-synthetic datasets. As far as the toy problem is concerned, I am slightly confused as to the choice of baseline, both in the regard to its training procedure and as to why this was deemed more suitable than one purposed for few-shot learning, so that we might go beyond simple verification to getting some initial sense for the performance of SIB. Moreover, it is not clear from the description as to how $\lambda$ is implemented here. As it stands, Section 5, for me, offers little in the way of valuable insights. The experiments section on the whole, results aside, feels somewhat rushed; the synthetic gradients being a potential limiting factor for instance feels "tacked on" and seems to warrant more than just a passing comment.

Minor errors
- Page 7: the "to" in "let $p_\psi(w)$ to be a Gaussian" is extraneous
- Page 8: "split" not "splitted".
- Further down on the same page, "scale" in "scale each feature dimension" should be singular and Drop" is misspelled as "dropp".
- Page 9: "We report results **with using** learning rate..."
- _Equation 17_ includes the indicator function $k \neq i$ but $i$ is not defined within the context.

EDIT: changed score

**Experience Assessment:**

I have read many papers in this area.

**Review Assessment: Checking Correctness Of Derivations And Theory:**

I carefully checked the derivations and theory.

**Review Assessment: Checking Correctness Of Experiments:**

I carefully checked the experiments.

**Review Assessment: Thoroughness In Paper Reading:**

I read the paper at least twice and used my best judgement in assessing the paper.

---

> ### Author Response · Authors · 2019-11-15
> **Thank you for a thorough review! (I/II)**
>
> Dear R2,
>
> Thank you for your detailed comments. We understand that the main issues that lead your vote to “weak reject” are:
> (a) Coherence in the presentation, especially when it comes to explaining motivation for transduction in an empirical Bayes setting.
> (b) Novelty.
> (c) You find the experiments “impressive” but would like to see more insights out of them.
>
> Regarding (a), we have addressed all of your concerns and argued about the importance of theoretically justifying transduction within Empirical Bayes; we explained how this is placed into context with regards to other frameworks that could support transduction. Importantly, improved generalization is evident from our motivation.
>
> Regarding (b), we argue that novelty comes from considering transduction in meta-learning, which improves generalization and, hence, performance.
>
> Regarding (c) we have now added the suggested experiments.
>
> We hope these changes and answers can make you consider raising your score.
>
> Detailed answers follow below.
>
> Q1. “The paper does, however, at times feel to be disjointed and, to an extent, lacking in focus.”
>
> Answer: We have improved the paper, and have made the story more coherent. See answers below.
>
> Q2. “it seems there is nothing prohibiting the substitution of the gradient for a learned surrogated just as well under a deterministic meta-initialization framework. As such, despite sporting impressive results on the image datasets, I am not convinced about how truly novel the method is when viewed as a whole.”
>
> Answer: Regarding the particular implementation of the transduction, we agree there will always be alternative ways to enable the transductive setting in a deterministic framework. However, we believe our approach is a particularly sensible one, because: (a) it respects the structure of the problem by emulating the optimization process at test time and (b) it incorporates both the support set (gradient steps) and the query set (initialization network) in an efficient manner that follows readily from a Bayesian theoretical derivation.
>
> We believe the novelty of considering a transductive setup holds independently of the Bayesian framework. However, the Bayesian framework gives us additional advantages and justification.
>
>
> Q3. “Theorem 1, it does feel a little out of place within the main text, especially since it is not specific to the transductive setting considered, nor even to the meta-learning setting more broadly”
>
> Answer: We have significantly reworked on section 4 and thus Theorem 1 to elaborate
> a) why transduction helps achieving good generalization
> b) the connection between empirical Bayes (EB) and information bottleneck leading to a generalization bound for EB models.
>
> The theorem is related to meta-learning since it answers many questions of EB, and EB is a good model for meta-learning.
>
> Please read the paragraphs "Implications of (14)" and "Implications of (15)" for detailed discussions of Theorem 1.
>
> We wish to clarify the importance of Theorem 1: it justifies theoretically the empirical Bayes formulation for meta-learning, which is a key element in our approach. As far as we know, this is the first such justification; indeed, previous theoretical analyses (e.g. Amir & Meir 2018) are not specialized to empirical Bayes.
>
>
> Q4. “more experiments, per Appendix C, highlighting the importance of transduction and therein the synthetic gradients and its formulation would be welcome.”
>
> Answer: We believe that Table 1 provides a strong evidence to show that transduction and synthetic stochastic gradient descent (SSGD) are important: compare the test accuracy between K=0 (without SSGD) with K=5 (three steps of SSGD) -- the improvement is about 10%.
>
> We have added an unsupervised domain adaptation experiment in Section 5.3 and show that synthetic gradient is important there as well.
>
> We have also added an ablation study on varying $n$ to check the influence on the testing accuracy.  The results are somehow inline with Theorem 1.
>
>
> Q5. “it is stated that an additional loss for training the synthetic gradient network to mimic the true gradient is unnecessary; while I agree with this conclusion, I likewise do not think it would hurt to explore use of the more explicit formulation.”
>
> Answer: To justify this intuition we compared to a variant which incorporates the gradient difference as an additional loss, on MiniImageNet with 1-shot, K=3, WRN-28-10. The test accuracy was 62.935%, which is about 6% lower than the accuracy (69.6%) reported in our Table 1. Therefore, we did not include this variant as a baseline.

---

> > ### Author Response · Authors · 2019-11-15
> > **Thank you for a thorough review! (II/II)**
> >
> >
> > Q6. “Considering the authors argue specifically for the importance of transduction in the zero-shot learning regime, I think it would be reasonable to expect experiments substantiating this, and the strength of their method in this regard, on non-synthetic datasets.”
> >
> > Answer: Following your suggestion and to increase convincingness, we have added an additional experiment on unsupervised domain adaptation in Section 5, which can be considered as zero-shot learning in the target domain. For more details, please take a look at Sec 5.3.
> >
> > Q7. “As far as the toy problem is concerned, I am slightly confused as to the choice of baseline, both in the regard to its training procedure and as to why this was deemed more suitable than one purposed for few-shot learning, so that we might go beyond simple verification to getting some initial sense for the performance of SIB. Moreover, it is not clear from the description as to how \lambda is implemented here. As it stands, Section 5, for me, offers little in the way of valuable insights. The experiments section on the whole, results aside, feels somewhat rushed; the synthetic gradients being a potential limiting factor for instance feels "tacked on" and seems to warrant more than just a passing comment.”
> >
> > Answer: The purpose of Section 5 (now Section 5.2) was to validate whether SIB can be applied to zero-shot learning, that is, without resorting to the support set.
> >
> > We agree that comparing to a BNN baseline is unfair in this case, and thus remove the comparison completely.  We have also rewritten Section 5.2 to clarify all the details.
> >
> > We would like to argue that Figure 3 offers an initial sense for the performance of SIB on zero-shot learning. Inspired by the success in toy data, we have conducted a new experiment of zero-shot classification on real data, which can be found in Section 5.3. Besides, we have also reorganized the experiments to make the presentation more fluent.

---

### Official Review · AnonReviewer1 · 2019-10-24
**Official Blind Review #1**

**Rating:** 6

**Review:**

The authors propose a method for transductive few-shot learning. The method is derived by taking a Bayesian perspective and recasting meta-learning as amortized variational inference, showing that results in a transductive scheme, and then using maml-style approximation of the inference (i.e., based on truncated stochastic gradient). While the idea of the paper seems intuitive, I find the writing quite confusing throughout (see my comments below) and I believe it must be improved before publishing the paper. Regarding the empirical evaluation, the results on standard benchmarks (miniImageNet and CIFAR-FS) seem reasonably strong; however, I would not call it "significantly outperform previous state-of-the-art" (as the authors claim in the abstract), since really all the top methods are in the same ballpark (the provided 95% CI overlap).


Comments:

1. In Eq. 2, if the task-specific losses are arbitrary, the whole construction is no longer a log-likelihood but rather just a loss. The authors also denote the distribution over the meta-training datasets as p_\psi(d_t), where d_t includes both inputs and targets. However, the concrete instantiations of the framework use discriminative models. Adjusting and clarifying the notation would improve the paper.

2. The way KL divergence is used in Eq. 5 is misleading since the arguments are two distributions over different sets of random variables. I would recommend keeping expected log conditional probability as a separate term (which is common in the literature).

3. Relatedly, going from ELBO to amortized VI (Eqs. 4-6) is a standard widely used VAE trick, so the derivation itself is not that informative. On the other, it would be great to include the inductive inference scheme mentioned right before Eq. 7 and compare it side-by-side with the standard amortized VI (Eq. 6). The way that part is presented now leaves the reader to derive all the details on their own.

4. Sec. 3, paragraph 1: While the original neural processes tend to underfit the data as pointed by the authors, more recent versions of the model such as attentive neural processes might work well, and perhaps worth mentioning.

5. Difference between Eq. 7 and 8 -- I believe I am misunderstanding this, but the updates look identical to me up to KL between q_\theta and a prior p_\psi. How exactly does \phi(x_t) parametrize the optimization process? I don't see how it enters into the equations. Generally, I feel deriving the method through a Bayesian perspective is quite confusing (as it is presented now) and way less clear than what is illustrated in Figure 1c.

6. Re: theoretical analysis -- it seems like the more than half a page spent on defining what generalization error is in the given setup (where all the definitions are quite standard), but then the discussion of the result, discussion of specific cases, connection to the information bottleneck bounds are all compressed down to in 1-2 sentences. This makes the "analysis" section really useless. Exemplifying the result of Thm. 1 and significantly elaborating the discussion would improve the paper.


Minor:

- The paragraph before Theorem 1: "Proposition" -> "Theorem"
- [UPD] Figure 3: titles, labels, ticks are all too small to be readable.

---------

Thanks to the authors for a detailed response. Most of my points have been addressed satisfactorily. I've updated my review accordingly.

**Experience Assessment:**

I have published one or two papers in this area.

**Review Assessment: Checking Correctness Of Derivations And Theory:**

I carefully checked the derivations and theory.

**Review Assessment: Checking Correctness Of Experiments:**

I carefully checked the experiments.

**Review Assessment: Thoroughness In Paper Reading:**

I read the paper at least twice and used my best judgement in assessing the paper.

---

> ### Author Response · Authors · 2019-11-15
> **Thank you for a thorough review!**
>
> Dear R1,
>
> Thank you for your comments. You cite presentation and experiments as the main reasons for “weak reject”. As you can see in the details below, we have now addressed all of your comments (and those of other reviewers) regarding presentation and have clarified the contribution of the experiments (including presenting new experiments). We hope this will make you consider raising your score.
>
> Detailed answers follow.
>
> “0. the empirical evaluation, the results on standard benchmarks (miniImageNet and CIFAR-FS) seem reasonably strong; however, I would not call it "significantly outperform previous state-of-the-art" (as the authors claim in the abstract), since really all the top methods are in the same ballpark (the provided 95% CI overlap).”
> Answer:
> We have now rephrased the results’ description using more conservative language. In Section 5, we have made clear that this is a method for small number of shots, where our results significantly outperform other methods. For example, in MiniImageNet 1-shot with WRN-28-10 backbone, SIB has a 5.9% improvement over the best previous results:
> CTM (Li et al. 2019): 64.1 +- 0.8%
> SIB (K=5): 70.0 +- 0.6%
>
> For 5-shot learning, SIB is not the best but is still competitive, as seen from Table 1:
> CTM (Li et al. 2019): 80.5 +- 0.1%
> Gidaris et al. 2019: 79.9 +- 0.3%
> SIB (K=5): 79.2 +- 0.4%
> Note that we did not use engineering tricks to improve our results. More discussions can be found in the end of Sec 5.1.
>
> “1. In Eq. 2, if the task-specific losses are arbitrary, the whole construction is no longer a log-likelihood but rather just a loss… $d_t$ includes both inputs and targets. However, the concrete instantiations of the framework use discriminative models. Adjusting and clarifying the notation would improve the paper.”
>
> Answer: We have adjusted the notation such that $\ell_t$ specifically denotes the discriminative model; we also removed the implication that arbitrary losses could be used (they could, in principle, but we do not consider this and we do not want to complicate the notation unnecessarily).
>
> “2. The way KL divergence is used in Eq. 5 is misleading”
>
> Answer: We agree and have changed accordingly to follow the standard notation in variational inference.
>
> “3. it would be great to include the inductive inference scheme mentioned right before Eq. 7 and compare it side-by-side with the standard amortized VI (Eq. 6).”
>
> Answer: In fact, there is no standard amortized VI in the case when both x and y are present. We argued that MAML amortizes the variational posterior as $q_{\phi(d_t^l)}$ while SIB uses $q_{\phi(x_t)}$ or $q_{\phi(x_t, d_t^l)}$. Since SIB makes use of unlabeled $x_t$, we call it transductive inference. We have now made a clear distinction/comparison between inductive inference and transductive inference in Section 2.2.
>
> “4. more recent versions of the model such as attentive neural processes might work well, and perhaps worth mentioning.”
>
> Answer: We have added some comments on ANP: it is powerful but in theory it takes O(n^2) time while SIB remains O(n) time complexity as the original NP.
>
> “5. Difference between Eq. 7 and 8 -- I believe I am misunderstanding this, but the updates look identical to me up to KL between q_\theta and a prior p_\psi. How exactly does \phi(x_t) parametrize the optimization process?”
>
> Answer: The main difference: eq.(7) (now eq.(6) in the revised version) operates on the support set $d_t^l$ while eq.(8) operates on the query set $d_t$; note that eq.(8) is the ideal case but we don’t have access to $y_t$, so we are not able to compute the gradient in eq.(8) unless we synthesize it.
>
> Regarding the parameterization of $\theta_t^K = \phi(x_t)$, it is basically a neural network in the sense that $\phi$ defines the mapping between $x_t$ and $\theta_t^K$. The mapping involves the synthetic gradient steps. We have added Figure 2 in the paper to illustrate the network architecture of $\phi(x_t)$.
>
> “6. RE Theorem 1, [...] then the discussion of the results, discussion of specific cases, connection to the information bottleneck bounds are all compressed down to in 1-2 sentences. This makes the "analysis" section really useless. Exemplifying the result of Thm. 1 and significantly elaborating the discussion would improve the paper.“
>
> Answer: We have significantly reworked on section 4 and thus Theorem 1 to elaborate
> a) why transduction helps achieving good generalization
> b) the connection between empirical Bayes (EB) and information bottleneck leading to a generalization bound for EB models.
>
> Please read the paragraphs "Implications of (14)" and "Implications of (15)" for detailed discussions of Theorem 1.
>
> We wish to clarify the importance of Theorem 1: it justifies theoretically the empirical Bayes formulation for meta-learning, which is a key element in our approach. As far as we know, this is the first such justification; indeed, previous theoretical analyses (e.g. Amir & Meir 2018) are not specialized to empirical Bayes.

---

### Official Review · AnonReviewer4 · 2019-10-31
**Official Blind Review #4**

**Rating:** 6

**Review:**

This paper addresses the issue of meta-learning in a transductive learning setting. That is, it aims to learn a model from multiple tasks and make it generalise to a new task in order to solve it efficiently. In the transductive setting, the query set (i.e., containing the unlabeled test data) of the new task is taken into account when learning the model.

This paper takes the empirical Bayes approach to meta-learning. In order to utilise the test data that do not access to groundtruth labels, it proposes to use synthetic gradient to implement the tranductive learning. A multi-layer perceptron network is used to systhesize the gradient. Theoretical analysis is conducted to demonstrate the generalization capability of the proposed model and reveal its connection to the information bottleneck principle in the literature of neural networks.

Overall, this is a well organised and nicely presented work. The idea on how to utilise the unlabeled test data to realise tranductive learning is novel; the analysis is thorough; and experimental study is provided to show the effectiveness of the proposed method. Meanwhile, this work can address the following issues:

1. The first paragraph on page 5, which describes the key step of syntheising gradient, can be made clearer;
2. In the experimental study, Table 1 compares various methods with the proposed one. It will be helpful to clearly indicate for each method in comparison whether/how it also utilises the query set. This will give more context in interpreting the comparison results;
3. The advantage of the proposed method seems to diminish quickly from 1-shot to 5-shot settings. Does this mean in the case of 5 (or more)-shot setting, considering unlabeled test data with the proposed method could even adversely affect the meta-learning performance? Please comment.
4. Since the proposed method works in a tranductive manner, it is presumed that the whole model needs to be retrained/updated once a new set of query data (e.g., for the same task or another new task) is given? In other words, how does the trained model generalise to unseen unlabeled test data? Please provide some discussion on this issue.
5. Finally, how is the computational complexity of training the proposed EB model?



**Experience Assessment:**

I have published one or two papers in this area.

**Review Assessment: Checking Correctness Of Derivations And Theory:**

I assessed the sensibility of the derivations and theory.

**Review Assessment: Checking Correctness Of Experiments:**

I carefully checked the experiments.

**Review Assessment: Thoroughness In Paper Reading:**

I read the paper at least twice and used my best judgement in assessing the paper.

---

> ### Author Response · Authors · 2019-11-15
> **Thank you for a thorough review!**
>
> Dear R4,
>
> Thank you for your insightful comments. Below we address one by one the issues that you mentioned.
>
> “1. The first paragraph on page 5, which describes the key step of syntheising gradient, can be made clearer”
>
> Answer: We agree and have added Figure 2 to illustrate the idea of the amortization using synthetic gradients.
>
>
> “2. In the experimental study, Table 1 compares various methods with the proposed one. It will be helpful to clearly indicate for each method in comparison whether/how it also utilises the query set. This will give more context in interpreting the comparison results;”
>
> Answer: Following the suggestions, we have added a clarification in Section 5.1 on this:
> “Apart from SIB, TPN (Liu et al., 2018) and CTM (Li et al., 2019) are also transductive methods.”
> Because Table 1 is already quite dense, we think this way may also be acceptable.
>
>
> “3. The advantage of the proposed method seems to diminish quickly from 1-shot to 5-shot settings. Does this mean in the case of 5 (or more)-shot setting, considering unlabeled test data with the proposed method could even adversely affect the meta-learning performance? Please comment.”
>
> Answer: We break down the two main arguments included in this comment.
> - Argument 1: “Advantage diminishes with larger number of shots”.
> Reply:
> We agree that our method's advantage is larger with smaller shots.
> We have clarified throughout the document (including the intro/motivation) that this is the setting where our method is better applicable to. We believe this does not diminish the significance of our method, it only specifies the scenarios where it is most useful.
>
> - Argument 2: “considering unlabeled test data with the proposed method could even adversely affect the meta-learning performance?”
> Reply:
> With cosine-similarity based classifier, regardless of 1-shot or 5 (or more)-shot, seeing additional points in the feature space should help us to sketch the distribution of features, and thus help the fast adaptation of the weights of the classifier. CTM (Li et al. 2019) was also motivated by this intuition.
> However, it doesn't mean the more unlabeled data the better. As argued by Theorem 1 (see the paragraph "Implications of (14)"), the meta-model (gradient network $\xi$ in our case) may not be able to absorb the amount of information efficiently resulting an over-regularization: note that there is a trade-off between the generalization error and the training error; when considering too many unlabeled data, we put a large weight on the generalization error.
> We have empirically confirm this on Mini-ImageNet. See Table 4 in the updated paper.
>
>
> “4. Since the proposed method works in a tranductive manner, it is presumed that the whole model needs to be retrained/updated once a new set of query data (e.g., for the same task or another new task) is given? In other words, how does the trained model generalise to unseen unlabeled test data? Please provide some discussion on this issue.”
>
> Answer: This situation is exactly what we mentioned “as in semi-supervised learning, an inductive learner can always be built from a transductive one”
> As long as we assume the test data are drawn from the same distribution, seeing some unlabeled points will only help us identify the distribution. We may choose to incorporate more unlabeled data to improve the model or we stop at some point and being inductive later on.
> An interesting discussion on this topic can be found here: http://olivier.chapelle.cc/ssl-book/discussion.pdf
>
>
> “5. Finally, how is the computational complexity of training the proposed EB model?”
> Answer:
> In theory, SIB has O(n) time complexity, where n is the number of examples of a task. In practice, for training SIB with WRN-28-10 backbone on a GTX Titan X GPU, it takes about 7 hours.

---

### Author Response · Authors · 2019-11-15
**Thank you for your comments**

Dear reviewers,

Thank you for your comments.

We reply to your comments with individual replies to your reviews, please see corresponding replies below. We believe we have addressed all of your concerns.

With this top-level comment we'd like to also highlight that:
- we have added code for our paper
- we have uploaded a new pdf which incorporates our edits according to your comments.

Many thanks,
Authors

---

### Decision · Program_Chairs · 2019-12-19

**Decision:**

Accept (Poster)

**Comment:**

Three reviewers have assessed this paper and they have scored it 6/6/6 after rebuttal. Nonetheless, the reviewers have raised a number of criticisms and the authors are encouraged to resolve them for the camera-ready submission.

---

> ### Author Response · Authors · 2020-02-15
> **Paper revised**
>
> Dear program chairs,
>
> We have revised our paper according to reviewer's comments and have made our code public for reproducing our few-shot learning experiments.